# Plant-Growth-Promoting Effect by Cell Components of Purple Non-Sulfur Photosynthetic Bacteria

**DOI:** 10.3390/microorganisms10040771

**Published:** 2022-04-02

**Authors:** Shuhei Hayashi, Yasunari Iwamoto, Yuki Hirakawa, Koichi Mori, Naoki Yamada, Takaaki Maki, Shinjiro Yamamoto, Hitoshi Miyasaka

**Affiliations:** 1Department of Applied Life Science, Sojo University, 4-22-1 Ikeda, Nishiku, Kumamoto 860-0082, Japan; ping.pondush@gmail.com (Y.I.); maruheta.p.d.leaf@softbank.ne.jp (Y.H.); kouichi09418@gmail.com (K.M.); syamamot@life.sojo-u.ac.jp (S.Y.); miyasaka@life.sojo-u.ac.jp (H.M.); 2Matsumoto Institute of Microorganisms Co., Ltd., 2904 Niimura, Matsumoto, Nagano 390-1241, Japan; yamada@matsumoto-biken.co.jp (N.Y.); maki@matsumoto-biken.co.jp (T.M.)

**Keywords:** purple non-sulfur photosynthetic bacteria, *Rhodobacter sphaeroides*, plant growth promotion, lipopolysaccharide

## Abstract

*Rhodobacter sphaeroides*, a purple non-sulfur photosynthetic bacterium (PNSB), was disrupted by sonication and fractionated by centrifugation into the supernatant and pellet. The effects of the supernatant and pellet on plant growth were examined using *Brassica rapa* var. *perviridis* (komatsuna) in the pot experiments. Both fractions showed growth-promoting effects: the supernatant at high concentrations (1 × 10^7^ to 4 × 10^7^ cfu-equivalent mL^−1^) and the pellet at a low concentration of 2 × 10^3^ cfu-equivalent mL^−1^). We expected lipopolysaccharide (LPS) to be the active principle of the pellet fraction and examined the effects of LPS on the growth of *B. rapa* var. *perviridis*. The growth of the plants was significantly enhanced by the foliar feeding of *R. sphaeroides* LPS at concentrations ranging from 10 to 100 pg mL^−1^. The present study is the first report indicating that LPS acts as one of the active principles of the plant-growth-promoting effect of PNSB.

## 1. Introduction

Microbial biomass presence and/or activity in the soil and rhizosphere can offer advantages at the level of organic matter deposition, stress mitigation, and environmental impact. Several mechanisms utilized by plant-growth-promoting bacteria have been discussed and considered [1,2,3]. Purple non-sulfur bacteria (PNSB), a plant-growth-promoting bacterium, is a phototrophic microorganism that has gained increasing attention in plant production due to its ability to produce and accumulate high-value compounds that are beneficial for plant growth [4]. PNSB are facultative anaerobic bacteria that can derive their energy from light, with their carbon source being derived from organic carbon, and are widely distributed in a variety of environments (e.g., moist soils, freshwater, and seawater) [5,6,7]. Furthermore, PNSB can execute a variety of useful functions such as nitrogen (N_2_) fixation [6,8], phosphate solubilization [1,9,10], and production of plant-growth-promoting substances (indole-3-acetic acid (IAA) and 5-aminolevulinic acid (ALA)) [7,11,12]. They have gained increasing attention in plant production because of their ability to benefit plant growth.

When used on plants, three categories of PNSB products are distinguished: living cells, inactivated cells, and cell-derived products. The existing literature mainly focuses on the use of living PNSB cells and less on the use of inactivated cells, while information regarding the use of extracted plant-growth-promoting substrates is scarce [4]. Living PNSB cells can be applied via the soil by either direct application or irrigation with a suspension containing the cells. Additionally, plant seeds can be coated with living PNSB biomass, providing direct contact with plant-growth-promoting substrates produced by the living cells to the plant at an early growth stage [1,13,14,15]. PNSB products with inactivated cells contain either autoclaved cells or dried biomass produced through freeze-drying, spray-drying, or heat-drying. Plant-growth-promoting substances might be contained in cellular components that do not sufficiently release other components by autoclaving or drying.

In this study, the inactivated cells were prepared by sonication to sufficiently release cellular components. The effects of plant growth promotion by dead and live cells were also investigated. Furthermore, we examined the plant-growth-promoting substrates contained in the inactivated cell components.

## 2. Materials and Methods

### 2.1. Bacteria, Plant and Chemicals

The PNSB *Rhodobacter sphaeroides* owned by the Matsumoto Institute of Microorganisms Co., Ltd. (Matsumoto, Japan), were used. The 16S rRNA sequence of this strain was identical to that of *R. sphaeroides* ATCC 17025 (GenBank accession number CP000661) [16]. *R. sphaeroides* was stored in a 20% glycerol solution at −80 °C. Regarding the colony forming unit (cfu), the correlation between the optical density at 660 nm and the cfu was obtained experimentally and standardized as 2 × 10^9^ cfu mL^−1^ when OD_660_ = 1. Komatsuna (*Brassica rapa* var. *perviridis*) seeds were purchased from Atariya Noen Co., Ltd. (Katori, Japan). Bacto yeast extract was purchased from Becton, Dickinson and Company (Sparks, MD, USA). Lipopolysaccharides of *R. sphaeroides* (LPS-RS) were purchased from InvivoGen (San Diego, CA, USA). Lipopolysaccharides of *Escherichia coli* O26 (LPS-EC) were purchased from FUJIFILM Wako Pure Chemical Corporation (Osaka, Japan). All other reagents were purchased from Nacalai Tesque, Inc. (Kyoto, Japan).

### 2.2. Preparation of Inactivated Cells and Cell Components

*R. sphaeroides* was inoculated into a GM medium (3.8 g L^−1^ L-glutamic acid, 2.7 g L^−1^ disodium DL-malate, 2.0 g L^−1^ Bacto Yeast Extract, 0.8 g L^−1^ (NH_4_)_2_SO_4_, 0.5 g L^−1^ KH_2_PO_4_, 0.5 g L^−1^ K_2_HPO_4_·2H_2_O, 0.2 g L^−1^ MgSO_4_·7H_2_O, 53 mg L^−1^ CaCl_2_·2H_2_O, 5 mg L^−1^ biotin, 5 mg L^−1^ nicotinic acid, 5 mg L^−1^ thiamine hydrochloride, and 1.2 mg L^−1^ MnSO_4_·5H_2_O) and cultured at 30 °C, 80 rpm, and under light irradiation for 3 days. The culture was centrifuged at 3500 rpm for 10 min at room temperature, and the supernatant was discarded. The pellet was resuspended in sterile water and centrifuged at 3500 rpm at room temperature for 10 min. The pellet was resuspended in sterile water to adjust the number of bacterial cells to 10^10^ cfu mL^−1^. The microbial suspension was used as the live-cell solution. The live-cell solution was sonicated using an ultrasonic device (TAITEC, VP-60s) four times for 30 s, with an interval of 30 s. The cell lysate from sonication was used as the inactivated cell solution. The inactivated cell solution was centrifuged at 15,560× *g* at room temperature for 10 min (Figure 1). The supernatant was transferred to a sample tube and used as the supernatant (soluble cell component, 1 × 10^10^ cfu-equivalent mL^−1^). The pellet was suspended in an equal volume of sterile water as the supernatant and used as a pellet solution (insoluble cell component, 1 × 10^10^ cfu-equivalent mL^−1^). 

### 2.3. Live or Inactivated Cells Experimental Design

The cultivating soil (Protoleaf (Tokyo, Japan)) was placed on a planter (length, 56 cm; width, 17 cm; height, 15 cm; Iris Ohyama, 650E) and provided sufficient water. Eighteen komatsuna seeds were sown per planter. Each planter was sprayed with 500 mL of live or inactivated cells diluted to 2 × 10^6^ cfu mL^−1^ every three days. On the day when the live or dead cell solutions were not sprayed, 500 mL of water was sprayed. Plant cultivation was carried out outdoors at Sojo University, mainly from September to November (the average temperature ranges from 15 °C to 25 °C). Approximately 30 days after the start of cultivation, the plants were harvested, and the fresh weight per plant without roots was measured.

### 2.4. Cell Supernatant or Pellet Experimental Design

The cultivating soil was placed on the planter and given sufficient water. Four komatsuna seeds were sowed per planter. Each plant was sprayed with 1 L per planter of the supernatant or pellet solution diluted to a certain concentration (the supernatant: from 2 × 10^6^ cfu-equivalent mL^−1^ to 4 × 10^7^ cfu-equivalent mL^−1^; the pellet solution: from 2 × 10^3^ cfu-equivalent mL^−1^ to 2 × 10^7^ cfu-equivalent mL^−1^) every three days. On the day when the supernatant or pellet solution was not sprayed, 1 L of water was sprayed. Plant cultivation was carried out outdoors at Sojo University, mainly from September to November (the average temperature ranges from 15 °C to 25 °C). Approximately 30 to 40 days after the start of cultivation, the plants were harvested, and the fresh weights per plant without roots were measured.

### 2.5. LPS Experimental Design

The cultivating soil was placed in pots (volume, 1.5 L; diameter, 16 cm; height, 14.5 cm) and provided with sufficient water. Komatsuna seeds were sown in each pot. LPS-RS solution (10 pg mL^−1^, 100 pg mL^−1^, and 1000 pg mL^−1^) and LPS-EC solution (100 pg mL^−1^) were sprayed into 100 mL pots every three days. On the day when the LPS solution was not sprayed, 100 mL of water was sprayed. Each condition was sprayed onto five pots. Plant cultivation was carried out outdoors at Sojo University, mainly from September to November (the average temperature ranges from 15 °C to 25 °C). Approximately 30 to 40 days after the start of cultivation, the plants were harvested, and the fresh weight per pot without roots was measured.

### 2.6. Statistical Analysis

We used two-tailed *t*-tests (* *p* < 0.05, ** *p* < 0.01) in order to compare different treatments with each other. All the biostatistical analysis was performed using Microsoft Excel office 365 data analysis toolkit.

## 3. Results

### 3.1. Effects of Soluble and Insoluble Fractions of Inactivated Cells on Plant Growth

Most of the previous studies focused on the effects of live PNSB cells on plant growth, but there have been a small number of studies reporting the effects of dead (inactivated) PNSB cells [4]. First, to confirm the growth-promoting effects of dead PNSB cells on plant growth, we examined the effects of whole inactivated cell fraction of *Rhodobacter sphaeroides*. The plants were treated with live and inactivated cells at a concentration of 2 × 10^6^ cfu mL^−1^ every 3 days by foliar feeding. Figure 2 shows the fresh weight of the plants (aerial part) after 30 d of cultivation. Control plants were sprayed with water instead of the cell suspension. The fresh weights were significantly (*p* < 0.01) increased in both live and dead cells treated plants respectively, and the dead cells showed a higher growth-promoting effect than the live cells.

All previous studies on dead PNSB cells examined the effects of the whole fraction of dead cells, but we expected that the soluble and insoluble fractions might act differently to plants, and fractionated the dead cells by centrifugation into the supernatant and pellet (Figure 1). We examined the effects of these two fractions on plant growth. Figure 3 shows the plants on day 30, and Figure 4 shows the fresh weight of the plants (aerial part) after 30 days of cultivation. For the supernatant (Figure 3A and Figure 4A), the growth of the plants was promoted at higher concentrations (1 × 10^7^ to 4 × 10^7^ cfu-equivalent mL^−1^), while for the pellet solution (Figure 3B and Figure 4B), growth was inhibited at higher concentrations (2 × 10^6^ to 2 × 10^7^ cfu-equivalent mL^−1^), and growth promotion was observed at a low concentration of 2 × 10^3^ cfu-equivalent mL^−1^.

### 3.2. Effects of Lipopolysaccharide (LPS) on Plant Growth

The well-known active principles of PNSB are IAA and ALA [7,11,12], and these low-molecular-weight compounds must be present in the supernatant fraction. In this study, we focused on the effects of pellets because there have been no reports on the growth-promoting effects of the insoluble fraction of dead PNSB cells. The pellet fractions showed a growth-promoting effect at a concentration of 2 × 10^3^ cfu-equivalent mL^−1^. The dry weight of *R. sphaeroides* was 5 mg/10^9^ cfu, thus 2 × 10^3^ cfu can be calculated as 2 ng of dry weight. This means that the active principle in the pellet fraction must have a plant-growth-promoting effect at the pg mL^−1^ to ng mL^−1^ order of concentration. *R. sphaeroides*, which is PNSB, is a Gram-negative bacterium, and general Gram-negative bacteria contain LPS as a cell membrane component and peptidoglycan (PGN) as a cell wall component; their physiological activities have been widely reported [17].

We hypothesized that LPS might promote the growth of plants; therefore, we conducted an experiment to spray LPS to plants. Figure 5 shows the fresh weights of the above-ground part of komatsuna when komatsuna plants were cultivated for 30 days by spraying LPS of *R. sphaeroides* (LPS-RS) or LPS of *E. coli* (LPS-EC) at various concentrations every 3 days. Komatsuna given LPS-RS at 10 pg mL^−1^ or 100 pg mL^−1^ had a significantly higher fresh weight than the control komatsuna (*p* < 0.01), indicating that growth was promoted. On the other hand, LPS-EC at the same concentration as 100 pg mL^−1^, which is the most effective concentration for LPS-RS, had the same fresh weight as the control komatsuna. So, the effect of LPS on plant growth was considered to be an effect peculiar to LPS-RS. Experiments were also performed in which LPS-EC was sprayed at 10 or 1000 pg mL^−1^, but the effect was not as good as when 100 pg mL^−1^ was sprayed (data not shown).

## 4. Discussion

Although there have been several studies reporting the plant-growth-promoting effects of inactivated PNSB [4], all the previous studies examined the effects of the whole dead cell fraction. In this study, we fractionated the dead cells into soluble (supernatant) and insoluble (pellet) fractions by centrifugation and separately examined the effects of each fraction on plant growth. The effective concentration of each fraction for plant growth was completely different, and growth promotion was observed at high concentrations in the supernatant and at low concentrations in the pellet. We also observed an inhibitory effect of pellets at high concentrations (2 × 10^6^ to 2 × 10^7^ cfu-equivalent mL^−1^).

Since the insoluble fraction (pellet) promoted plant growth at an extremely low concentration, LPS—which is a cell wall component contained in the pellet—was predicted to be a component of plant-growth-promoting activity. When LPS derived from PNSB was sprayed on komatsuna and the growth-promoting effect was investigated, a significant promoting effect was confirmed at a low concentration of 100 pg mL^−1^. The effective concentration of LPS (10 to 100 pg mL^−1^) was in good accordance with the effect of the pellet, as shown in Figure 4B. Growth promotion by pellets was observed at a concentration of 2 × 10^3^ cfu-equivalent mL^−1^. Our data show that the dry weight of 10^9^ cells was approximately 5 mg (data not shown). The amount of LPS is reportedly 5–10% of the cell dry weight [18]. The concentration of LPS in the pellet at 2 × 10^3^ cfu-equivalent mL^−1^ was calculated using the following formula: (2 × 10^3^/10^9^) × 5 mg × (0.05 to 0.1) = 2 × 5 ng × (0.05 to 0.1) = 1000 pg to 2000 pg mL^−1^.

Since LPS from *E. coli* did not show a plant-growth-promoting effect at the same concentration (100 pg mL^−1^) (Figure 5), the plant-growth-promoting effect of LPS was considered to be characteristic of PNSB. This was considered to be due to the difference in the structure of LPS between the two. The structure of lipid A of LPS from *E. coli* contains six fatty acyl groups, whereas LPS from PNSB contains five fatty acyl groups [19]. Therefore, LPS from PNSB is a potent antagonist of toxic LPS in both human and murine cells and prevents LPS-induced shock in mice. LPS from PNSB appears to utilize at least two distinct mechanisms to block LPS-dependent activation of Toll-like receptor 4 (TLR4). The main mechanism involves direct competition between under-acylated LPS and hexa-acylated LPS for the same binding site on myeloid differentiation protein 2 (MD-2), while the secondary mechanism involves the ability of under-acylated LPS/MD-2 complexes to inhibit hexa-acylated endotoxin/MD-2 complexes to function at TLR4 [20,21,22,23]. Differences in plant-growth-promoting effects might also be attributed to differences in fatty acyl groups.

The plant-growth-promotion effect of LPS has been reported so far on wheat (*Triticum aestivum*) treated with *Azospirillum brasilense* Sp245 LPS, but its effective concentration was 100 mg mL^−1^ [24]. LPS from PNSB in the present study was effective at 100 pg mL^−1^, indicating that the LPS of PNSB had extremely high plant-growth-promoting activity. To date, plants have been considered less sensitive to LPS than animals. However, the concentration of 100 pg mL^−1^, which showed the effect of promoting plant growth in this study, was the same as that of animal cells, which is the first report of plant cells responding at the same concentration as animal cells.

## 5. Conclusions

There have been many studies reporting the plant-growth-promoting effects of dead PNSB cells, but all the previous studies examined the effects of the whole fraction of dead cells. In the present study, we fractionated the inactivated cells by centrifugation into the supernatant and pellet fractions, and found that the supernatant and pellet showed different concentration ranges in growth-promoting effects (supernatant at a higher and pellet at a lower concentration). Based on this finding, we expected the lipopolysaccharide (LPS) in pellet fraction would serve as the active principle, and showed that LPS from *R. sphaeroides* at 100 pg mL^−1^ had a remarkable plant-growth-promoting effect. The present study is the first report indicating that LPS acts as one of the active principles of the plant-growth-promoting effect of PNSB. In addition, plants have generally been believed to be much less sensitive to LPS compared to animals, and the concentration of LPS required for plant growth-promotion were reportedly 10 to 100 μg mL^−1^. The present study is also the first report indicating that LPS acts on plants at the concentration as low as they do in animals. With respect to practical applications, the LPS from PNSB must be a promising candidate as a growth-promoting substance for agriculture.

## Figures and Tables

**Figure 1 microorganisms-10-00771-f001:**
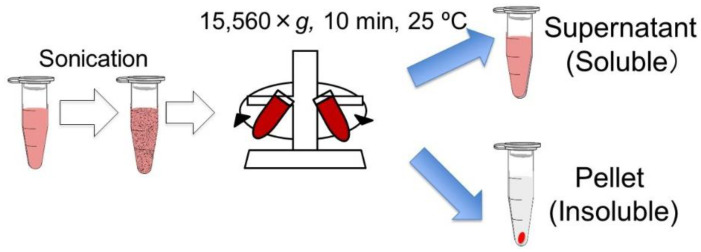
Scheme of preparation of inactivated cells and cell components.

**Figure 2 microorganisms-10-00771-f002:**
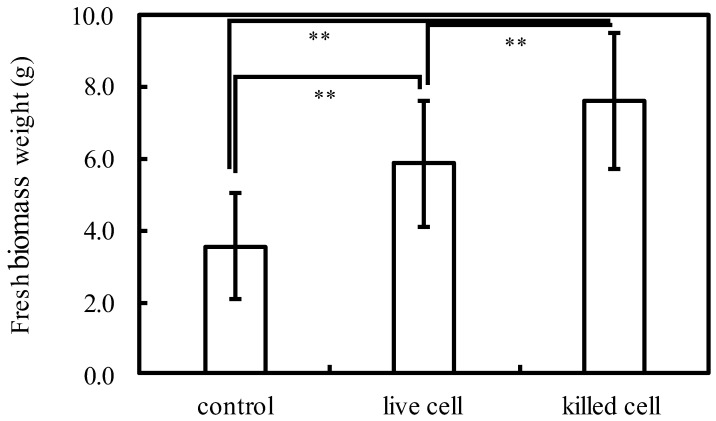
Fresh weight of komatsuna when live or inactivated PNSB (2 × 10^6^ cfu mL^−1^) were sprayed once every 3 days and cultivated for 30 days. Each bar represents the mean ± S.D. (*n* = 18). Asterisks indicate significant difference at *p* < 0.01 (**).

**Figure 3 microorganisms-10-00771-f003:**
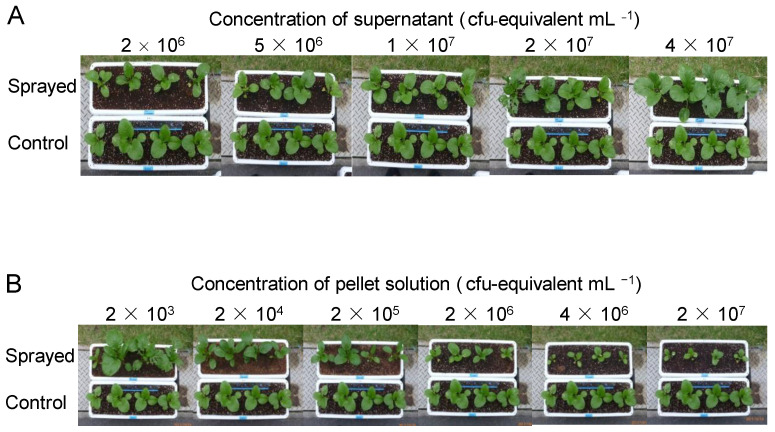
Plants (komatsuna) cultivated for 30 days. (**A**) Upper: plants sprayed with supernatant at each concentration. (**B**) Upper: plants sprayed with pellet solution at each concentration. (**A**,**B**) Lower: control (without supernatant or pellet solution).

**Figure 4 microorganisms-10-00771-f004:**
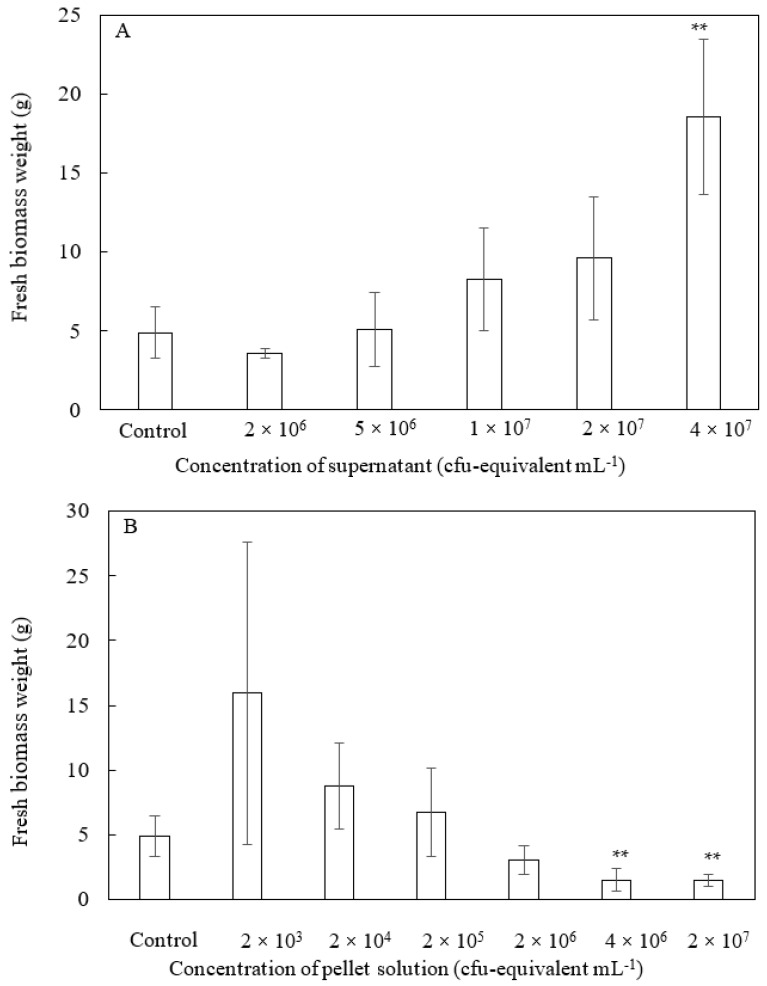
Fresh weight of komatsuna when supernatant (**A**) or pellet solution (**B**) was sprayed once every 3 days and cultivated for 30 days. Each bar represents the mean ± S.D. (*n* = 4). Asterisks indicate significant difference with control at *p* < 0.01 (**).

**Figure 5 microorganisms-10-00771-f005:**
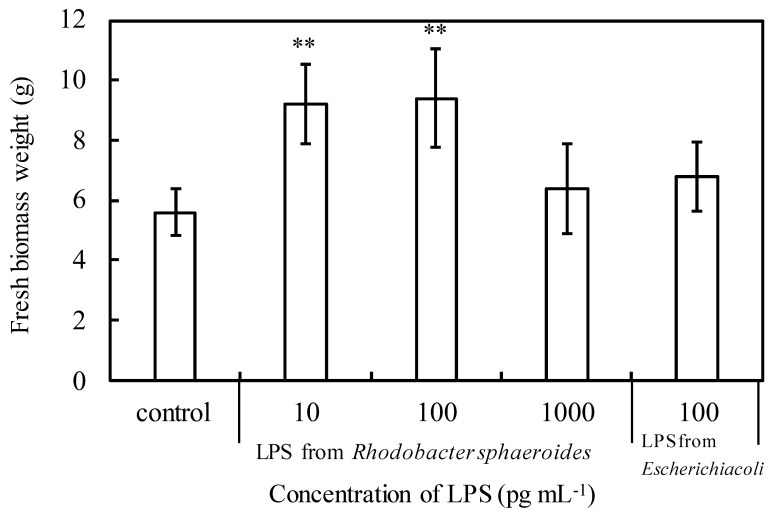
Fresh weight of komatsuna when LPS solutions were sprayed once every 3 days and cultivated for 30 days. Each bar represents the mean ± S.D. (*n* = 5). Asterisks indicate significant difference with control at *p* < 0.01 (**).

## Data Availability

The data presented in this study are available upon request from the corresponding author.

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
