# Peer review of "Plant-Growth-Promoting Effect by Cell Components of Purple Non-Sulfur Photosynthetic Bacteria"

_microorganisms, 2022, doi:10.3390/microorganisms10040771_

Round 1
Reviewer 1 Report
The paper presents the results obtained in vitro, by using the products obtained from the inactive cell of Rhodobacter sphaeroides on the growth of the plants’ Brassica rapa var Perviridis. All experiments were performed in pots.
The biological material obtained from the microorganism Rhodobacter sphaeroides was inactivated by ultrasound. The resulting mixture was separated by centrifugation and divided into pellets (insoluble part made up of cell walls) and supernatant (the soluble part inside the cell). These two bioproducts have been standardized, as CFU/mL.
Both bioproducts were used as liquid fertilizers for the plants. Results obtained reveal that the bioproducts based on a suspension of pellets from Rhodobacter sphaeroides (insoluble cells walls) stimulate the plant growth at the concentration of 2 x 103 CFU echivalent•mL-1. Bioproducts obtained from the supernatant (soluble part from bacterial cell) stimulate the plant growth at a concentration of 4 x 107 CFU equivalent •mL-1.
Regarding the results obtained from the bioproduct which contain insoluble parts of cells (cells wall in the fact), these are compared with results obtained by application of pure LPS from E. coli. the authors concluded that the superior results obtained in the case of Rhodobacter sphaeroides are specific to specific lipopolysaccharides containing this microorganism.
Now the paper is clear are well written; however, minor corrections or additional information are needed, regarding the following aspects:
1) The sentence ‘’killed cells’’ is not appropriate for a scientific paper; it must be replaced with the sentence ‘’inactivated cells’’
2) The authors must be mentioned the mode of standardization of their samples in the chapter named ‘’Materials and Methods’’
(I suppose that the solutions of pellets and supernatant resulting after sonication are standardized as’’ UFC/mL via measurements of turbidity or optical density (OD), taking into account the McFarland relationship between OD and CFU?)
Author Response
Thank you for reviewing our manuscript and pointing out minor corrections.
1) The sentence ‘’killed cells’’ is not appropriate for a scientific paper; it must be replaced with the sentence ‘’inactivated cells’’
We replased all "killed cells" to "inactivated cells".
2) The authors must be mentioned the mode of standardization of their samples in the chapter named ‘’Materials and Methods’’ (I suppose that the solutions of pellets and supernatant resulting after sonication are standardized as’’ UFC/mL via measurements of turbidity or optical density (OD), taking into account the McFarland relationship between OD and CFU?)
It was closer to what the reviewer thought, but we did our own experiments to find out the correlation between optical density and colony forming units. So we added the following sentence to Chapter 2.1.
Regarding the colony forming unit (cfu), the correlation between the optical density at 660 nm and cfu was obtained experimentally and standardized as 2 × 109 cfu mL-1 when OD660 = 1.
Also, for pellet solution and supernatant, we revised as follows to make standardization easier to understand in Chapter 2.2.
The supernatant was transferred to a sample tube and used as the supernatant (soluble cell component, 1 × 1010 cfu-equivalent mL-1). The pellet was suspended in an equal volume of sterile water as the supernatant and used as a pellet solution (insoluble cell component, 1 × 1010 cfu-equivalent mL-1).
Reviewer 2 Report
In this paper, the authors tested the capacity of the lipopolysaccharides from Rhodobacter sphaeroides to promote plant growth after the observation that dead cells promote the growth of komatsuna more than the alive ones.
It is an interesting way to use bacteria as fertilizers because with this kind of inoculation the different problems associated with the bacteria are avoided.
Seems like this manuscript have been reviewed before and all the corrections improve the read and understanding of the manuscript.
However, I suggest another few corrections very simples:
Line 37: remove the space between the parenthesis and the word indole.
Line 66: The sentence "R. sphaeroides was stored in a 20% glycerol solution at −80°C." should be transferred to section 2.1.
Line 66: After the previous sentence, R. sphaeroides is missing before "was inoculated".
Line 224: I think that the conclusion is written like a summary of the results. Conclusions should be more direct. Modify it, please.
Line 227: Brassica rapa and perviridis should be in italic and perviridis without a capital letter.
Figure 5: Escherichia are wrog written.
With the suggested changes, I propose to accept the manuscript in this journal because this study provides novelty to the journal.
Author Response
Thank you for reviewing our manuscript and pointing out minor corrections.
Line 37: remove the space between the parenthesis and the word indole.
We deleted the space.
Line 66: The sentence "R. sphaeroides was stored in a 20% glycerol solution at −80°C." should be transferred to section 2.1.
We reprinted "R. sphaeroides was stored in a 20% glycerol solution at −80°C." in section 2.1.
Line 66: After the previous sentence, R. sphaeroides is missing before "was inoculated".
We added “R. sphaeroides” in the sentence.
Line 224: I think that the conclusion is written like a summary of the results. Conclusions should be more direct. Modify it, please.
We revised Conclusion more directly.
Line 227: Brassica rapa and perviridis should be in italic and perviridis without a capital letter.
Due to the revision of the Conclusion, the relevant part had been deleted.
Figure 5: Escherichia are wrog written.
We fixed it.
This manuscript is a resubmission of an earlier submission. The following is a list of the peer review reports and author responses from that submission.
Round 1
Reviewer 1 Report
It is recommended to reject the manuscript. The reasons are as follows :
1、The title of this paper is mainly about the growth promoting effect of lipolysaccharide in bacteria, but lipolysaccharide wad not described in the Introduction, only Rhodobacter sphaeroides (a purple non-sulfur photosynthetic bacterium) had been introduced. Therefore, the relationship between lipolysaccharide and Rhodobacter sphaeroides is not clear at all. Moreover, most of the content of the paper is about the effects of the supernatant and pellet on plant growth and the effects of plant growth promotion by dead and live cells, which is inconsistent with the title.
2、Many details in the paper are unclear, for example, (1)Statistical methods should be added to material methods.(2)Plant growth conditions should be added in line 90-91.(3)The specific concentration shall be specified in line 94-95. (4)In Figure 5, What concentration does ‘100 (LPS-EC)’ represent in X axis?
3、There is no direct relationship between lipolysaccharide and Rhodobacter sphaeroides.
Reviewer 2 Report
The authors presented in brief their results obtained with a bioproduct resulting from non-sulfur photosynthetic bacteria Rhodobacter sphaeroides, (PNSB) on the plant species Brassica rapa, in pots.
The bioproducts used in experiments were: suspensions of living bacteria, suspensions of inactivated bacteria, and the bioproducts obtained from bacterial disruption by sonication ( i.e. supernatants and pellets). Results obtained are interpreted in comparison with untreated plants and respectively by plant treatment with two commercials lipopolysaccharides, one derived from Escherichia coli and second derived from PNSB.
The obtained results indicate that the supernatants and the pellets have growth-promoting plant growth effects as follows: the supernatant obtained from solutions that contain initially high concentrations of bacteria (1 × 107 to 4 × 107 CFU equivalent /mL), and the pellets obtained from solutions that contain initially at a low concentration of bacteria ( 2 × 103 CFU equivalent /mL).
In my opinion, results presented by authors in their manuscript are promising, but minor revisions are needed before publishing, as follow :
1) In chapter 2.3: In each pot were sown 18 Brassica rapa seeds? From the picture presented in Fig. 3 A and B in each pot appear to be sown only 4 seeds. Authors must explain this in the text.
2) To support affirmation according to that the growth of the plants was significantly enhanced by the foliar feeding of LPS obtained from Rhodobacter spheroides, at concentrations ranging from 10 to 100 pg mL-1, in figure 5 authors must present results obtained by using both commercial lipopolysaccharides ( i.e. the influence of concentrations LPS from Escherichia coli and LPS from Rhodobacter sphaeroides at concentrations of 10 pg/mL, 100 pg/mL and 1000pg/mL on quantities of fresh biomass obtained;
3) on the axis Oy from Figure 2, from Figure 4, and from figure 5 which will be revised, ''Fresh weight (g)'' must be replaced by ''Fresh biomass weight (g)'';
4)The authors must introduce in their article a Chapter named Conclusions.